# A Reinforcement Learning Handover Parameter Adaptation Method Based on LSTM-Aided Digital Twin for UDN

**DOI:** 10.3390/s23042191

**Published:** 2023-02-15

**Authors:** Jiao He, Tianqi Xiang, Yixin Wang, Huiyuan Ruan, Xin Zhang

**Affiliations:** School of Information and Communication Engineering, Beijing University of Posts and Telecommunications, Beijing 100876, China

**Keywords:** ultra-dense network, handover parameters, deep Q-learning, digital twin, long short-term memory

## Abstract

Adaptation of handover parameters in ultra-dense networks has always been one of the key issues in optimizing network performance. Aiming at the optimization goal of effective handover ratio, this paper proposes a deep Q-learning (DQN) method that dynamically selects handover parameters according to wireless signal fading conditions. This approach seeks good backward compatibility. In order to enhance the efficiency and performance of the DQN method, Long Short Term Memory (LSTM) is used to build a digital twin and assist the DQN algorithm to achieve a more efficient search. Simulation experiments prove that the enhanced method has a faster convergence speed than the ordinary DQN method, and at the same time, achieves an average effective handover ratio increase of 2.7%. Moreover, in different wireless signal fading intervals, the method proposed in this paper has achieved better performance.

## 1. Introduction

In recent years, the number of mobile users around the world has exploded, and so has the data traffic. A study found that the number of mobile terminal devices worldwide could reach tens of billions by 2023 [1]. At the same time, the rapid development of the mobile internet has also promoted the rapid popularization of mobile communication. As an important entrance to the mobile internet, the penetration rate of smart terminals has grown exponentially, which has greatly stimulated the development of the mobile internet and accelerated global users’ demands for wireless data services. As the demand for wireless data services grows, existing network deployments cannot support the explosive growth of data services. In this context, Ultra-Dense Networks (UDN) technology, which increases the number of micropower base stations and reduces each base station’s coverage radius, is an effective and affordable solution [2].

Dense deployment of small base stations could shorten distances as well as greatly improve link quality between mobile stations and base stations, improve signal coverage, and increase system capacity [3]. However, due to the complexity of terminal movement and the small-scale characteristics of a single base station, the coverage of most micro base stations is between 50 and 200 m. With the development of wireless access technology, this distance could even be reduced to 10 m [4]. This inevitably leads to more frequent handover of terminals between base stations, resulting in unprecedented signaling overhead. At the same time, due to the rapid changes in signal strength of the low-power micro-base station transmissions received by mobile users, the probability of handover failure and ping-pong handover also increases significantly, resulting in data packet loss and delay that damage network performance and user experience. Therefore, how to design a reasonable handover strategy to reduce the number of unexpected events during the user’s mobility, such as ping-pong handover and handover failure, and ensure user experience has become an urgent problem to be solved [2].

At present, there have been a lot of research on cell handover, mainly by optimizing the trigger threshold and trigger time to improve the handover performance. The authors in [5] proposed a reduced early handover (REHO) scheme by adjusting the time to trigger (TTT) in the long-term evolution (LTE) system. The researchers in [6] presented a Markov-chain-based handover mechanism in the Fifth generation (5G) ultra-dense networks, which enables users to achieve early handover by predicting the next target base station. The authors in [7] proposed a deep Q-learning (DQN) algorithm to adaptively adjust the handover trigger time, based on the access rate and the signal to interference noise ratio (SINR). The authors in [8] proposed a long short term memory (LSTM)-based prediction method to find a proper handover point in advance through analyzing the historical reference signal receiving power (RSRP) and predicting the future changes of RSRP. The above studies mainly focus on optimizing the handover trigger point to reduce the handover failure rate and improve the stability of Quality of Service (QoS) during the handover process, but do not consider the resulting problem of ping-pong handover.

In order to better optimize the handover problem, the handover failure rate and the handover ping-pong rate need to be considered jointly. The authors in [9] proposed a frequent handover mitigation algorithm to reduce the handover failure rate and ping-pang rate in UDN. The suggested algorithm monitors the serving cell history and the dwelling time of the users to detect fast mobile users or ping-pong users. Simulation results indicated that the suggested algorithm can reduce 79.56% of the overall handovers at the cost of high complexity. The authors in [10,11] proposed several handover parameter optimization algorithms to overcome ping-pang problems, but only for the one-dimensional optimization of the hysteresis value (Hys). Furthermore, the authors in [12] proposed that Hys and TTT are not independent, and provided a functional relationship between Hys, TTT, and UE velocity, which can dynamically select reasonable handover parameters without the need for any predefined handover failure or ping-pang handover thresholds. The researchers in [13] simulated the effect of different handover parameter settings on 5G network performance at different speeds and proposed that medium handover parameter settings are the best solution. However, this paper analyzed fixed handover parameter settings so the algorithm could not optimize parameters automatically.

Based on joint optimization, system adaptive optimization is an important research direction to dynamically achieve the optimal performance in the actual communication process. The authors in [14] proposed a Q-learning-based algorithm in LTE network to dynamically adjust Hys and TTT. The authors in [15] constructed a parameter adaptive handover mechanism suitable for 5G in the high-speed railway dedicated LTE-R communication system, the mechanism uses the interaction of temporal-difference (TD)-learning-based reinforced agents to obtain high-speed railway handover and network performance in different combinations of speeds and handover parameters. The simulation results indicated that the suggested mechanism can find the optimal handover parameters to improve the handover performance and network performance. The researchers in [16] proposed an automatic weighting function to adjust the weights of different functions according to the SINR level of the user equipment (UE), the cell load and the speed of the UE, thereby independently adjusting the handover parameter values for the users. On the basis of [7], the authors in [17] proposed a DRQN algorithm to dynamically adjust Hys and TTT, considering ping-pong handover and handover failure. However, the above research have the following disadvantages: for the flexibility and performance of the algorithm, the protocol stack has been modified to a large extent which, for example, embeds the DQN into the existing handover process, resulting in frequent Q-network invoking every time when facing potential handover. Mobile devices that do not support software definition of communication protocols are also unable to deploy those algorithms and thus fall back to legacy handover process. This is considered to be low in efficiency, may contain poor backward compatibility, and is difficult to be implemented.

The motivation of this paper is to establish a handover adaptive optimization method with strong backward compatibility. This paper regards reinforcement learning module as an independent module in relation to the handover process. Thus, the information interaction between the two needs to be reduced as much as possible. Therefore, in this paper, signal fading conditions are regarded as the basis for optimization. Furthermore, in this paper’s context, the optimization module is not supposed to participate in the handover process but only gives parameter suggestions according to signal fading conditions.

This paper proposes a DQN handover parameter optimization scheme based on LSTM-assisted digital twins. A DQN handover parameter selection method for different wireless signal fading conditions is established, an LSTM-assisted digital twin is utilized to enhance performance by improving system efficiency and convergence effect. The main contributions of this paper are summarized as follows:Considering different conditions of signal fading, a real-time handover parameter selection method using a DQN is proposed. The method can be added to the actual communication system as an additional module without changing the handover process and has better backward compatibility. Good convergence performance is obtained through the validation of simulation platform.An enhanced DQN method based on digital twin is proposed. RSRP data that can trigger different TTT parameters in the actual system are collected, and used as the input of digital twins to predict the reward value under the assumed handover parameters. The virtual rewards from digital twin act on the DQN together with the actual system reward values. Compared with the ordinary DQN, it has faster convergence speed, and the final convergence effect is also better and more stable.Digital twin with LSTM network as the core is established. The RSRP temporal series between UE and each base station before triggering the handover event is used as LSTM input to predict whether the handover failure and ping-pong occur. Unbiased estimation of handover failure rate and ping-pong rate based on this method is proposed. With the aid of LSTM-based digital twin, the enhanced DQN achieves 2.7% higher effective handover rate than the ordinary DQN, and 10.9% higher than random parameter strategy.

The following paper is organized as follows. Section 2 introduces the system model and the optimization problem considered in this paper. Section 3 introduces the DQN handover parameter optimization algorithm to obtain the optimal strategy for selecting parameters according to signal fading conditions. Section 4 discusses the digital twin of the DQN, the enhanced method uses LSTM prediction as the core to build a digital twin. Section 5 gives the experimental results and analysis, and Section 6 concludes this paper.

## 2. System Model and the Optimization Problem

In a mobile communication network, in order to ensure the communication quality of the mobile terminal UE, instant handover to a base station with better link quality is an effective guarantee for its link quality. Especially with the increase of the communication frequency band, the coverage distance becomes smaller and the density of the base station is also increasing to ensure the coverage quality. The mobility management of UDN is also one of the key requirements for determining the system capacity. Usually handover events can be triggered by the relative signal strength relationship between the serving base station and neighboring base stations. For example, in the A3 event [18], which is the typical event for handover triggering, the following relationship needs to be satisfied to maintain TTT measurement periods:(1)$$RSRP_{n}>RSRP_{s}+\mathrm{Hys},$$
where Hys is an offset value, $RSRP_{n}$ is the reference signal receiving strength of a neighboring base station, and $RSRP_{s}$ is the reference signal receiving strength of the serving base station. In brief, A3 event requires that the signal strength of the neighboring base station exceeds a certain threshold Hys of the serving base station and that it is maintained for a certain period of time TTT. However, in an actual system, there are various phenomena that are not conducive to the stability and efficiency of the communication link, such as handover failure and handover ping-pong, which necessitate strict requirements on the setting of handover parameters.

Handover failure refers to the unexpected interruption of the handover process due to various reasons, such as weak link quality or strong interference [11]. The handover failure rate can be calculated as:(2)$$p_{f}=\frac{N_{HO\_failure}}{N_{HO}},$$
where $N_{HO\_failure}$ is the number of handover failures and $N_{HO}$ is the total number of handovers. Usually, more conservative handover parameters such as a larger TTT and a larger Hys could lead to handover failure more easily because conservative parameters tends to miss the best handover timing. However, in UDN, close inter-site distance makes the interference more severe, and unsuitable handover parameters could lead to a higher failure rate.

Handover ping-pong is an event where a UE is handed over to a new base station and then is handed over back to the original base station in less than a certain period of time [12]. The handover ping-pong rate can be calculated as:(3)$$p_{pp}=\frac{N_{HO\_pingpong}}{N_{HO}},$$
where $N_{HO\_pingpong}$ is the number of ping-pong handover. This phenomenon indicates that the current UE does not need to handover to the target base station, which can be regarded as a redundant handover and also a burden on the mobile communication system. Ping-pong handover increases signaling overhead for handover, and is not conducive to improving communication efficiency. Under normal circumstances, aggressive handover parameters tend to lead to more ping-pong handover, such as a smaller TTT and a smaller Hys, which could cause easier and more frequent triggering of A3 events [19], increasing the proportion of ping-pong handover in all handover. In UDN, the phenomenon of ping-pong handover are more obvious because the closer the station distance, the more drastic the signal strength change, and once the handover parameters are set unreasonably, the ping-pong rate would also rise sharply [17].

Neither handover failure or handover ping-pong is the expected effective handover. So the effective handover rate is defined as:(4)$$p_{valid}=1-p_{f}-p_{pp},$$
and the optimization problem in this paper is maximizing the effective handover rate through searching handover parameters, expressed as:(5)$$\begin{array}{c} \underset{w\in \Omega_{\mathrm{HO}}}{\max}p_{eff,\;w}\\ \begin{array}{cc} s.t. & \mathrm{Hys}>0\\ & \mathrm{TTT}>0\\ & p_{f}+p_{pp}<1 \end{array} \end{array},$$
where $p_{eff,\;w}$ is the effective handover rate with handover parameter. and $w\in \Omega_{\mathrm{HO}}=\left\{\left(\mathrm{Hys},\;\mathrm{TTT}\right)|\mathrm{Hys}\in \Omega_{\mathrm{Hys}},\;\mathrm{TTT}\in \Omega_{\mathrm{TTT}}\right\}$, where $\Omega_{\mathrm{HO}}$ denotes the set of all configurable handover parameters, $\Omega_{\mathrm{Hys}}$ for all Hys and $\Omega_{\mathrm{TTT}}$ for all TTT.

## 3. Handover Parameter Decision DQN Based on Signal Fading

Wireless signal fading is one of the important factors affecting communication systems and it also has a great impact on handover performance [20]. Under different signal fading conditions, the handover performance also has different performances. When signal fading is slight, the fluctuation of signal strength is small, and thus ping-pong handover is less likely to occur. At this time, lower Hys and TTT parameters, that is, more aggressive handover parameters, can be used to ensure a higher success rate. When the signal fading condition is severe, the signal strength fluctuates greatly, and thus, ping-pong handover is more likely to occur. It is necessary to adopt a more conservative strategy (larger Hys, TTT parameters) to reduce the ping-pong rate. Although a higher effective handover rate may be obtained, the handover failure rate might also increase [21]. Therefore, it is necessary to select different handover parameters for different fading conditions to ensure the robustness of the system [22].

Assuming that when the fading condition changes, which is described as fluctuation in both time and space domain in [20], the handover failure rate and ping-pong rate of the system also change accordingly, and the optimization problem presented in formula (5) is also to be transformed into the following optimization problem in order to get the optimal decision strategy $\hat{V}$:(6)$$\hat{V}=\underset{V}{\mathrm{argmax}}\int_{sd_{1}}^{sd_{2}}p_{valid,\;V\left(sd\right)}dsd,$$
where $sd\in \left[sd_{1},\;sd_{2}\right]$ is the fading standard deviation, V(sd) denotes the handover parameter decision function based on sd:(7)$$V:\left[sd_{1},\;sd_{2}\right]\to \Omega_{\mathrm{HO}},$$

Since the fading standard deviation is a continuous value, it is difficult for training samples to traverse all situations, so it is necessary to find a suitable decision modeling method. Compared with methods such as interpolation and polynomial fitting, the Q-learning method based on deep neural network is expected to have better results [14].

Although there is some research on the DQN method for handover parameter optimization, it has the following disadvantages: either the parameter selection is not optimized for signal fading conditions, or there are too many protocol procedures that need to be modified, resulting in poor backward compatibility. The purpose of this paper is to establish a DQN while considering signal fading characteristics with good backward compatibility. The DQN realizes the best action selection given by the Markov decision process (MDP) [14]. It models the surrounding environment as several states and actions, and agents use unsupervised learning to learn the surrounding environment. The agent makes actions that receives rewards to update current decision strategy. The purpose of the DQN is to find the optimal decision strategy, which utilizes the Q-network to perform the best action $\hat{a_{t}}$ based on current state $s_{t}$. In this paper, the signal fading condition is regarded as state $s\in \Omega_{s}$, where $\Omega_{s}=\left\{s|s\in \left[sd_{1},\;sd_{2}\right]\right\}$ is the state space. Furthermore, the handover parameter configuration is regarded as action $a\in \Omega_{a}$, where $\Omega_{a}$ represents the action space, which corresponds to all configurable handover parameters in this paper, in other words, $\mathrm{card}\left(\Omega_{a}\right)=\mathrm{card}\left(\Omega_{\mathrm{HO}}\right)$. As shown in Figure 1, the Q-network is established to obtain the value $Q\left(s_{t},\;a_{t}\right)$ of each action $a_{t}$ in the state $s_{t}$, and the action decision is determined by the following formula [23]:(8)$$\hat{a_{t}}=\underset{a_{t}\in \Omega_{a}}{\mathrm{argmax}}Q\left(s_{t},\;a_{t}\right),$$

In order to obtain a suitable Q-network, it is necessary to interact with the real environment and obtain the corresponding reward value of an action. In this paper, after the action $a_{t}$ has been made, the reward value at moment *t* + 1 is calculated as:(9)$$r_{t+1}=1-\alpha_{1}p_{f}-\alpha_{2}p_{pp}-\alpha_{3},$$
where $\alpha_{1}$, $\alpha_{2}$ and $\alpha_{3}$ are scaling coefficients. After getting the reward value, the weights of the Q-network could be updated according to the back propagation principle, where the loss value can be calculated by the following formula:(10)$$L={\left(Q\left(s_{t},\;\hat{a_{t}}\right)-r_{t+1}\right)}^{2}.$$

It should be noted that the DQN in this paper can be deployed on the terminal side in order to ensure that the difference of signal conditions of each UE is taken into account.

This paper now exhibits the basic DQN algorithm.
**Algorithm 1: DQN Algorithm**Initialize Q-network;Initialize ε=1;**for** (*i* = 1; *i* < nperiod; *i*++) **do** Update standard deviation of signal strength *sd*; **if** (i mod nε==0) **do**  ε←βε; **end if** **if** (U(0,1)<ε) **do**  Select an action randomly; **else do**  Predict an action with Q-network forward propagation; **end if** Set A3 parameters by the selected/predicted action; Collect HO failure rate *p_failure_* and HO pingpong rate *p_pingpong_*; Calculate reward *r* as in (9); Update Q-network with loss calculated as in (10);**end for**

Where $n_{\mathrm{period}}$ is the total number of training cycles, *sd* is the standard deviation of signal fading, which can be obtained by measuring resource blocks in the time-frequency domain, and ε is a random exploration factor. In each training cycle, based on the fading standard deviation real-time measurement, the best action is predicted by the Q-network. After corresponding this action to the A3 parameter, the handover failure rate and ping-pong rate of the system are collected, and the reward value is calculated. Then the reward value is utilized to update the Q-network. In each iteration, there is a probability ε to choose a random action instead of using the Q-network prediction. The purpose of this mechanism is to ensure the initial random exploration and reduce the chance of falling into a worse local optimum. At the same time, ε gradually decays with the iterative process, which becomes β times that of the previous one for each $n_{\varepsilon }$ iterations. This mechanism ensures the random decay in the initial phase, as well as stabilize the final convergence results of the Q-network.

The advantages of algorithm 1 in this paper are as follows. The signal fading condition is considered, specifically, when the signal fading is in a low range, the Q-network tends to adopt a more aggressive handover parameter strategy, or otherwise a conservative handover parameter strategy; this DQN module does not participate in the whole handover process, but just fetches the handover event statistics to give parameter suggestions, in other words, is independent from traditional mobile communication systems, which does not change the existing A3 handover process, resulting in strong backward compatibility.

Algorithm 1 also has the disadvantages of most DQN methods, that is, when the action space to be predicted is large, the convergence effect would degrade; the DQN in this paper utilizes continuous value of signal fading standard deviation as input, which puts forward higher requirements to the generalization ability of the Q-network. This may take a long time to train in order to achieve better prediction performance. In the following, an LSTM-based digital twin enhancement strategy is proposed to improve the convergence efficiency and performance of the DQN.

## 4. Enhanced DQN Based on Digital Twin

### 4.1. Digital Twin Enhancement Mechanism

Digital twin is a virtual representation of the physical world, which continuously interacts with the physical world and predicts and supplements various behaviors of the physical world [24]. The application of digital twins is often due to the limitations of observations in the physical world, and thus additional virtual modeling is required to supplement predictions. Through digital twin, additional information is obtained to improve the performance of physical systems. Deep learning is an important driver of digital twin, because deep learning itself is a data-driven model, which can obtain expected patterns based on real-world data feedback. As mentioned in [25], 5G networks and related applications often utilize rich data in different formats, based on which, deep learning can be used to create models of 5G traffic/network behavior with the knowledge of historical data as well as real-time Internet of Things (IoT) data from multiple sources to help detect anomalies and predict any potential bottlenecks in traffic performance. Deep learning-based digital twins can also significantly improve network performance in multiple domains and layers, including optimization of physical layer operations, power control, modulation and coding schemes, waveform selection, and mobility management based on signal-to-interference-plus-noise ratio prediction, caching in mobile edge computing, network slicing, etc. [26].

This paper also utilizes an Artificial Intelligence (AI) method to create a copy of the real world to help predict real-world handover events. In general, the real world is only able to provide limited feedback, which in this paper’s context, is the real-time rewards for the selected handover parameters. However, some data for other handover parameters, such as RSRP, could also be collected at the same time. Digital twin is proposed to utilize these data to predict rewards.

Specifically, after a communication system selects a set of A3 handover parameters, the handover behavior must satisfy the TTT window length. In other words, when a handover is triggered, the A3 buffer has retained the RSRP value of TTT consecutive measurement periods, which also includes potential A3 events that can be triggered with a lower TTT value, as shown in Figure 2. Furthermore, even if the A3 event corresponding to the current TTT value is not triggered, the A3 events corresponding to a lower TTT value may also exist. However, since the actual system can only perform handover according to one set of parameters, only the reward value corresponding to one set of parameters can be obtained, which is a waste of potential A3 event triggers.

This proposed digital twin is also considered to be deployed on the terminal side. This premise ensures that the approach in this paper is UE-specific, that is, data collection and processing, virtual event prediction as well as handover parameter decision are completed at the terminal side.

Through the digital twin based on the LSTM network, the potential A3 events could also be taken advantage of. Specifically, after the potential A3 event is triggered, the RSRPs of each base station at several historical-measurement periods are fed to the digital twin to obtain the handover failure rate and the ping-pong rate prediction for the corresponding handover parameter. Then, its reward value is calculated and fed back to the Q-network. The digital twin is able to take advantage of the RSRP values that would have been difficult to utilize and enrich rewards. The specific construction method of the digital twin is explained in Section 4.2.

The algorithm of Digital twin enhanced DQN (DTe-DQN) is as follows.
**Algorithm 2: DTe-DQN Algorithm**Initialize Q-network;Initialize ε=1;**for** (*i* = 1; *i*
$\leq n_{\mathrm{period}}$; *i*++) **do** Update standard deviation of signal strength *sd*; **if** ($i\;\mathrm{mod}\;n_{\varepsilon }==0$) **do**  ε←βε; **end if** **if** (U(0,1)<ε) **do**  Select an action randomly; **else do**  Predict an action with the Q-network forward propagation; **end if** Set A3 parameters $\left({\mathrm{Hys}}_{i},{\;\mathrm{TTT}}_{i}\right)$ by the selected/predicted action; Update digital twin A3 parameter set $\Omega_{\mathrm{DT}}=\left\{\left(\mathrm{Hys},\;\mathrm{TTT}\right)|\mathrm{Hys}={\mathrm{Hys}}_{i},\;\mathrm{TTT}<{\mathrm{TTT}}_{i},\;\mathrm{TTT}\in B\right\}$ Collect HO failure rate *p_failure_* and HO pingpong rate *p_pingpong_*; Calculate reward *r* as in (9); Update Q-network as in (10); **for** (*j* = 1; *j*
$\leq \mathrm{card}\left(\Omega_{\mathrm{DT}}\right)$; *j*++) **do**  Collect RSRP sequential values triggered by A3 parameter $\left({\mathrm{Hys}}_{\mathrm{DT},\;j},{\;\mathrm{TTT}}_{\mathrm{DT},\;j}\right)$;  Predict HO failure rate *p_failure_* and HO pingpong rate *p_pingpong_* via LSTM-based digital twin;  Calculate reward *r* as in (9);  Update Q-network with loss calculated as in (10); **end for** **end for**

Where $\Omega_{\mathrm{DT}}$ is a set containing all potential trigger parameters, which includes all parameters with Hys equal to the physical system and smaller TTT values, and $\left({\mathrm{Hys}}_{\mathrm{DT},\;j},{\;\mathrm{TTT}}_{\mathrm{DT},\;j}\right)$ is the *j*-th element in $\Omega_{\mathrm{DT}}$.

By predicting the virtual handover reward value from the digital twin, algorithm 2 is able to obtain more diverse reward information within a limited data collection period. The diversity of rewards is reflected that under the same signal fading condition, reward values for different handover parameters can be obtained. Diverse rewards can greatly improve the search efficiency in the DQN training process, thereby alleviating the convergence difficulties caused by large-scale action space and continuous value input. Note that if the real-world reward of a feedback cycle were repeatedly used to update the Q-network, the performance would not improve. This is because this does not introduce more effective exploration, but repeats old experience. Instead, the repeated training might lead to a worse local optimum.

It should also be noted that both the ordinary DQN algorithm and DTe-DQN algorithm have low requirements on the computational resources or efficiency of the actual system. In each iteration, statistics of several handover events are collected for the iterative process and the act of gathering itself might take a long time. Although this process could be accelerated by sharing handover information between multiple UEs, it is not possible to take less than the time it takes that most UEs perform a handover, i.e., in the order of seconds. Therefore, the efficiency of the algorithms is mainly limited by the real-time feedback rather than computational resources or power where the algorithms are deployed.

### 4.2. Digital Twin Based on LSTM

The LSTM-based digital twin proposed in this paper includes the following stages: real-world data collection, LSTM-network prediction, and unbiased estimation of handover failure rate and ping-pong rate, as shown in Figure 3.

In the first stage, the digital twin reads data from the A3 buffer. When some candidate base stations in the A3 buffer satisfy the Formula (1) for a certain length ${\mathrm{TTT}}_{\mathrm{DT},\;j}$, the RSRP of several measurement periods before this moment is extracted as the input of the LSTM network. Although the current actual handover TTT length is larger, these A3 triggers with smaller TTT are regarded as virtual handovers in digital twin. It should be noted that all potential A3 events triggered by this module are based on the same Hys value as the physical system, which can ensure better backward compatibility of the method, or otherwise additional A3 buffers might be needed.

In the second stage, the prediction task of the LSTM network is to predict whether the current handover is successful and whether ping-pong handover occurs according to the RSRP series. The RSRP series is expected to reflect the user’s relative signal strength trend with each base station, so machine learning can be used to extract possible future events. LSTM has the ability to process temporal series data [27]; the authors in [8] utilized LSTM to find the optimal handover timing by analyzing historical RSRP data and predicting future RSRP data. In [28,29], LSTM was used for wireless network traffic prediction.

LSTM is a special kind of recurrent neural network (RNN), mainly to alleviate the problem of gradient disappearance and gradient explosion during long sequence training [30]. Only one state exists within a single loop of an ordinary RNN, while four states exist within a single loop (also called a cell) of an LSTM. The cell state stores the state information of the current LSTM cell and passes it to the LSTM cell at the next moment. The current cell receives the cell state from the previous moment and works together with the signal input received by the current cell to generate the cell state of the current cell. In each cell, specially designed “gates” are used to introduce or remove information in the cell state. As shown in Figure 4, The LSTM cell is composed of an input gate, a forget gate, an output gate and a cell state: the input gate determines how much of the input data of the network at the current moment needs to be saved to the cell state; the forget gate determines how much of the unit state at the previous moment needs to be retained to the current moment; and the output gate controls how much of the current cell state needs to be left to the current output value [28].

The LSTM network in this paper is constructed as shown in Figure 5. The network includes an input layer, an LSTM layer, a fully-connected layer (FC layer) and an output layer. At each moment, the current RSRPs from the base stations considered are fed to the network. The output layer is a softmax layer for classification task of two classes, represented as “0/1”. The LSTM layer at h_t_ needs to receive information from the LSTM layer at h_t−1_ and deliver information to the LSTM layer at h_t+1_. The LSTM network is now in place, using historical RSRPs to predict whether handover failure or ping-pong occurs.

In the third stage, after establishing an LSTM network that predicts a single handover failure or handover ping-pong event, this paper proposes an unbiased estimation method for the failure rate and ping-pong rate in the digital twin. Taking handover failure as an example, if LSTM prediction is not deployed, the number of handover failures observed in actual system under the $n_{1}$ measurements is assumed to follow the Bernoulli distribution *X*$\sim ℬ\left(n_{1},p_{f}\right)$, where $p_{f}$ is the handover failure probability, let $S_{x}=\left\{x_{1},\;x_{2}I.,x_{n_{1}}\right\}$, $x_{i}\in \left\{0,\;1\right\}$ represents the sample of the *i*-th measurement. $x_{i}=1$ is for event occurring, i.e., the handover failing, and $x_{i}=0$ is for event not occurring. Then the maximum likelihood unbiased estimation of $p_{f}$ can be expressed as:(11)$$\hat{p_{f}}=\frac{1}{n_{1}}\sum_{i=1}^{n_{1}}x_{i}.$$

This is an estimation method when statistics from an actual system can be collected. However, when using LSTM to predict a single event, uncertainty would be introduced because its true positive probability $p_{p}$ and true negative probability $p_{n}$ are not 1. Here $p_{p}$ represents the probability that the LSTM predicts the event to occur when it does occur, and $p_{n}$ represents that the LSTM predicts the event not to occur when it does not occur. Therefore, the estimator (11) could not be applied for scenarios with LSTM prediction. After obtaining a series of event samples $S_{y}=\left\{y_{1},\;y_{2},\ldots ,y_{n_{2}}\right\}$, $y_{i}\in \left\{0,\;1\right\}$ predicted by LSTM, this paper proposes the following proposition.

Proposition 1:

With LSTM-predicting events, the unbiased estimation of $p_{f}$ is expressed as:(12)$$\hat{p_{f}}=\frac{p_{n}-1+\frac{1}{n_{2}}\sum_{i=1}^{n_{2}}y_{i}}{p_{n}+p_{p}-1}.$$

Proof.

If the handover failure were predicted to occur, it could be a correct prediction of the event occurring or a false prediction of the event not occurring. So the probability of LSTM predicted handover failure is calculated as:(13)$$\begin{array}{ll} p\left(y_{i}=1\right) & =p\left(x_{i}=1\right)p\left(y_{i}=1|x_{i}=1\right)+p\left(x_{i}=0\right)p\left(y_{i}=1|x_{i}=0\right)\\ & =p_{f}p_{p}+\left(1-p_{n}\right)\left(1-p_{f}\right) \end{array}.$$

The number of LSTM predicted failures obeys the Bernoulli distribution $Y\sim ℬ\left(n_{2},\;p_{Y}\right)$, $p_{Y}=p\left(y_{i}=1\right)=p_{f}p_{p}+\left(1-p_{n}\right)\left(1-p_{f}\right)$, where $p_{Y}$ is derived by Equation (13). Then, the unbiased estimate of the Bernoulli distribution parameter $p_{Y}$ can be obtained by the maximum likelihood criterion:(14)$$\hat{p_{Y}}=\frac{1}{n_{2}}\sum_{i=1}^{n_{2}}y_{i}.$$

Since estimator (14) is the unbiased estimator of $p_{Y}$, the expectation of estimator (14) is equal to $p_{Y}$, expressed as:(15)$$E\left(\frac{1}{n_{2}}\sum_{i=1}^{n_{2}}y_{i}\right)=p_{f}p_{p}+\left(1-p_{n}\right)\left(1-p_{f}\right).$$

Calculate the expectation of Formula (12) substituting Formula (15) and the following equation can be derived:(16)$$E\left(\hat{p_{f}}\right)=p_{f}.$$

Since the expectation of the estimator (12) for the parameter $p_{f}$ is equal to $p_{f}$, the estimator (12) is the unbiased estimation of $p_{f}$. Proposition 1 is proved. 

Corollary 1:

The variance of unbiased estimation of (12) is expressed as:(17)$$D\left(\hat{p_{f}}|S_{y}\right)=\frac{1}{n_{2}}\cdot {\left(\frac{1}{p_{n}+p_{p}-1}\right)}^{2}p_{Y}\left(1-p_{Y}\right).$$

Proof:

Due to Bernoulli distribution $Y\sim ℬ\left(n_{2},\;p_{Y}\right)$, the variance of unbiased estimation of $p_{Y}$ is expressed as:(18)$$\begin{array}{ll} D\left(\hat{p_{Y}}\right) & ={\left(\frac{1}{n_{2}}\right)}^{2}\cdot n_{2}p_{Y}\left(1-p_{Y}\right)\\ & =\frac{1}{n_{2}}p_{Y}\left(1-p_{Y}\right) \end{array}.$$

Get the variance of Formula (12) and substitute (18) to derive the following equation:(19)$$\begin{array}{ll} D\left(\hat{p_{f}}|S_{y}\right) & ={\left(\frac{1}{p_{n}+p_{p}-1}\right)}^{2}\cdot D\left(\frac{1}{n_{2}}\sum_{i=1}^{n_{2}}y_{i}\right)\\ & =\frac{1}{n_{2}}\cdot {\left(\frac{1}{p_{n}+p_{p}-1}\right)}^{2}p_{Y}\left(1-p_{Y}\right) \end{array}.$$

Corollary 1 is proved.

The variance of unbiased estimation in actual system is expressed as:(20)$$D\left(\hat{p_{f}}\right)=\frac{1}{n_{1}}p_{f}\left(1-p_{f}\right).$$

Compared with (20), corollary 1 indicates that after the prediction of LSTM, the prediction error of LSTM results in an increase in the estimation variance, thereby reducing the estimation performance. On the other hand, as the number of samples increases, the estimation variance is able to decrease steadily. In the LSTM-based digital twin proposed in this paper, the predicted handover failure rate and ping-pong rate correspond to more aggressive handover parameters with smaller TTT value, which brings much more potential triggers than the real physical-world system. Therefore, more sample data are available, compensating for the performance penalty introduced by LSTM. In general, the accuracy of unbiased estimation based on digital twin compared to that based on data collected in the physical world is mainly determined by the following variables: controllable $p_{n}$, $p_{p}$ and uncontrollable $n_{2}$. $p_{n}$ and $p_{p}$ could be promoted by adapting LSTM-network structure and training methods, and $n_{2}$ mainly depends on the UDN geometry.

Similarly, the unbiased estimation of handover ping-pong rate based on LSTM prediction can also be expressed as:(21)$$\hat{p_{pp}}=\frac{p_{n,\;pp}-1+\frac{1}{n_{2}}\sum_{i=1}^{n_{2}}y_{i}}{p_{n,\;pp}+p_{p,\;pp}-1},$$
where $p_{p,\;pp}$ and $p_{n,\;pp}$ are respectively the true positive probability and true negative probability of the LSTM network implementing ping-pong event prediction, and the variance of the unbiased estimate can also be organized into a form similar to formula (17).

Three steps are required to implement the digital twin. Firstly, a dataset needs to be collected for the LSTM network. The dataset should include historical RSRPs and whether handover failure or ping-pong occurred accordingly, which is collected in the actual communication network. The dataset should also be divided into a training set, a validation set and a test set. Second, the LSTM network is trained with the training and validation set and then evaluated with the test set. The true positive and true negative probabilities are derived. Thirdly, the digital twin is deployed. After the preparation of the previous two steps, the A3 buffer reading, LSTM real-time prediction and unbiased estimation are online and provide virtual rewards for handover parameter optimization.

## 5. Experiment Results and Analysis

A simulation platform has been built to illustrate the proposed method which realizes the air interface communication and handover process of the cellular system, as well as implements the two DQN algorithms proposed in this paper. Table 1 presents the assumption of some important parameters.

Data set for LSTM training and validating is first gathered. The true positive and true negative possibilities of LSTM-predicting events achieve 91.64% and 87.96% for handover failure and 92.19% and 91.98% for handover ping-pong, respectively. In addition, the number of samples entering the digital twin is at least 1.4 times that of the physical world. Thus, the unbiased estimation variance of the digital twin is at most 1.12 times (for failure prediction) and 1.01 times (for ping-pong prediction) that of the physical world. It can be considered that the estimation accuracy of digital twins is at the same level as that of the physical world.

Figure 6 shows the average reward iterative curves of the two DQN methods. It can be seen that no matter which method is applied, the reward can be gradually increased until it is roughly stable at a higher value, and the DTe-DQN method is better than the ordinary one in the following three aspects. First, the DTe-DQN has a faster convergence speed. This advantage reveals that in the early stage of algorithm iteration, more reward information can be collected quickly within a limited measurement feedback cycle. Both rewards from the real world or the digital twin makes the exploration more efficient. Second, the DTe-DQN has strong stability, which has less fluctuation and is more difficult to degenerate to a worse local optimum. This is still due to more exploration opportunities, guaranteeing robustness. Finally, the DTe-DQN has better performance and can get a higher reward value, which is the result of the combined effect of convergence efficiency and robustness.

Figure 7 shows the average handover failure rate and ping-pong rate during the iterative process. Both methods can achieve the decrease in the average handover failure rate and ping pong rate, but the DTe-DQN is more efficient and stable. Although the handover failure rate has only dropped by about 2–3%, the ping-pong rate has decreased by about 8–9%. It is revealed that in the scenario of UDN, ping-pong rate is an easier optimization indicator, or in other words, the final effective handover rate improvement depends more on the decline of ping-pong rate. The iterative curve of the average effective handover rate is presented in Figure 8. In addition to the fact that the effective handover rate of the DTe-DQN increases faster at the initial stage, the final convergence effect of the DTe-DQN is higher than that of the DQN. The former is about 83.1% and the latter is about 80.4%.

Since this paper is optimized for different signal fading conditions, the performances of different fading intervals are also the focus of this paper. Figure 9 shows the iterative process of the average reward in different fading intervals. The fading intervals here are divided into 1~2 dB, 2~3 dB, and 3~4 dB according to the fading standard deviation. It can be seen that as the signal fading condition becomes severe, the achievable reward declines, and the convergence effect fluctuates more. Therefore, the larger the signal fading standard deviation is, the harder the optimization is. Although there are differences in the reward value of different reward intervals, no matter which interval it is, the DTe-DQN is better than the DQN, reflecting the advantages mentioned earlier.

Figure 10 and Figure 11 are the iterative curves of the average ping-pong rate and the average failure rate of each fading interval. First of all, with the increase of fading standard deviation, there is a higher probability of ping-pong, reflecting higher optimization difficulty. In the 3~4 dB interval, the ordinary DQN method has declined to a worse local optimal solution. Even so, the DTe-DQN still has better performance. As for the handover failure rate, the higher the fading degree is, the smaller the improvement effect of the two methods could achieve. It can be derived that in the higher fading range, the optimization mainly focuses on reducing high ping-pong rate, whose benefit is far greater than reducing the failure rate, and this phenomenon is obvious in high fading interval.

We contribute this to the joint optimization reward and the generalization ability of the Q-network. This paper designs reward function with the aim of maximizing effective handover rate. Thus, the overall reward has been increased, even though there is little improvement in handover failure rate of high fading interval. Furthermore, the strong generalization ability of the Q-network enables digging out better parameter decision strategy on maximizing reward.

Figure 12 shows the iterative results of effective handover rate in different fading intervals. It can be seen that for 3~4 dB interval, the effective handover rate has been improved by about 16%. This is mainly because if the handover parameters were not optimized, the high signal fading would lead to a higher ping-pong probability. So its optimization space is significantly larger than 1~2 dB interval, about 7%. In addition, the DTe-DQN method also has a higher effective handover rate than the DQN most of the time. This figure validates the advantage of the DQN design based on signal fading that in each fading interval, both proposed algorithms have improved effective handover rate and achieved convergence. In other words, the proposed algorithms take advantage of the signal fading to improve handover performance, which would have degraded the handover performance due to fluctuation in time and space domain.

Figure 13 and Figure 14 summarize the handover ping-pong rate and failure rate that can be achieved in various cases. It can be seen that in relatively low fading interval, the handover failure rate and ping-pong rate have both been reduced at the same time. In high fading interval, however, the failure rate becomes harder to decrease, but the ping-pong rate is also decreased, making up for the shortcoming brought by failure rate, which can be validated from the comparison of the DQN and the DTe-DQN in 3~4 dB fading interval. This is due to the joint optimization design as discussed earlier that the handover failure and ping-pong rate are optimized for the overall effective handover rate.

Figure 15 summarizes the achievable effective handover rate in various cases. Whether it is average effective handover rate or in each fading interval, the DTe-DQN has 2~3% better performance than the DQN. In addition, the DTe-DQN performs 7~16% better than random handover parameter strategy, especially in high fading interval. It can be seen that the DTe-DQN proposed in this article can not only comprehensively improve the handover performance, but also outperforms the ordinary DQN in terms of high fading interval with a higher difficulty of optimization. We contribute it to the additional exploration mechanism of the DTe-DQN. The results of achievable handover performance reveals that with the aid of digital twin, the DTe-DQN achieves more effective interaction, which accelerates and stabilizes the DQN algorithm iteration.

## 6. Conclusions

In UDN, handover faces severe problems such as large interference and high ping -pong rate. This paper starts with the optimization problem of effective handover rate and proposes a dynamic DQN algorithm based on signal fading condition. This method has a good backward compatibility and can provide parameter optimization suggestions for the mobile communication system as an independent module. In order to improve the efficiency and performance of the ordinary DQN method, this paper proposes a digital-twin-enhanced DQN based on the LSTM network. Taking advantage of potential handover parameters before actual handover is triggered and corresponding RSRP sequences, handover failure rate, ping-pong rate, and corresponding reward value are estimated in LSTM-based digital twin. The proposed digital twin could increase the exploration efficiency of the DQN algorithm. The simulation results show that the two DQN methods mentioned in this article have significantly promoted effective handover rate, while the DTe-DQN has better convergence efficiency and robustness than the DQN, and eventually could achieve higher effective handover rate.

## Figures and Tables

**Figure 1 sensors-23-02191-f001:**
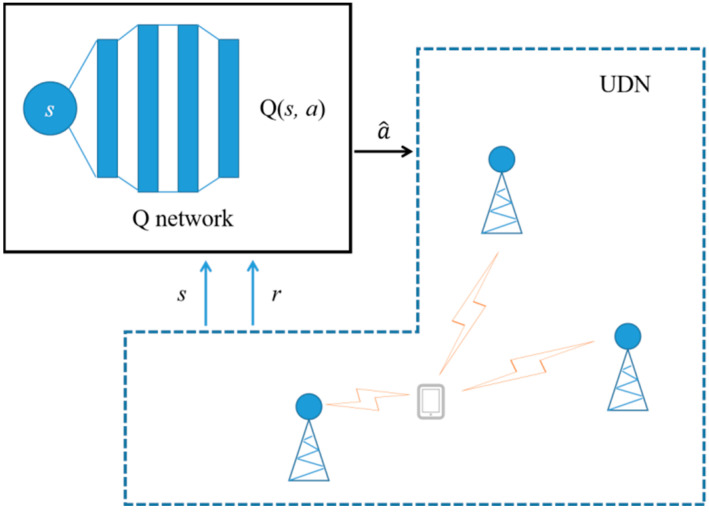
The DQN algorithm.

**Figure 2 sensors-23-02191-f002:**
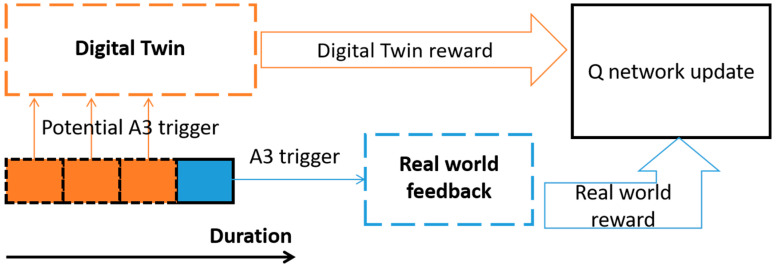
Digital twin enhanced DQN.

**Figure 3 sensors-23-02191-f003:**
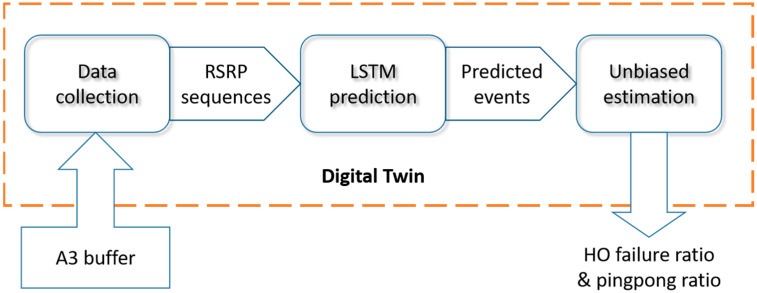
Stages of the DTe-DQN.

**Figure 4 sensors-23-02191-f004:**
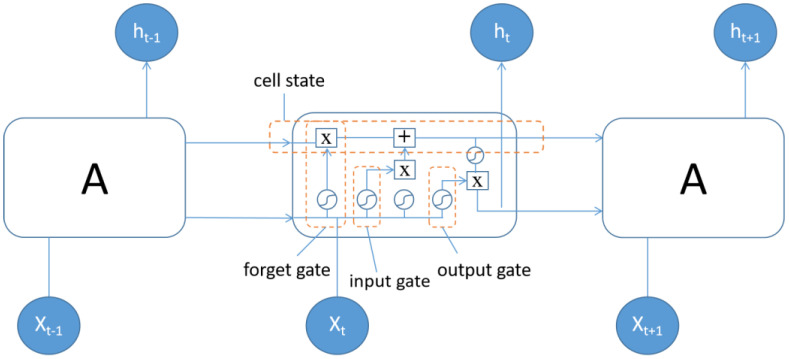
LSTM cell structure.

**Figure 5 sensors-23-02191-f005:**
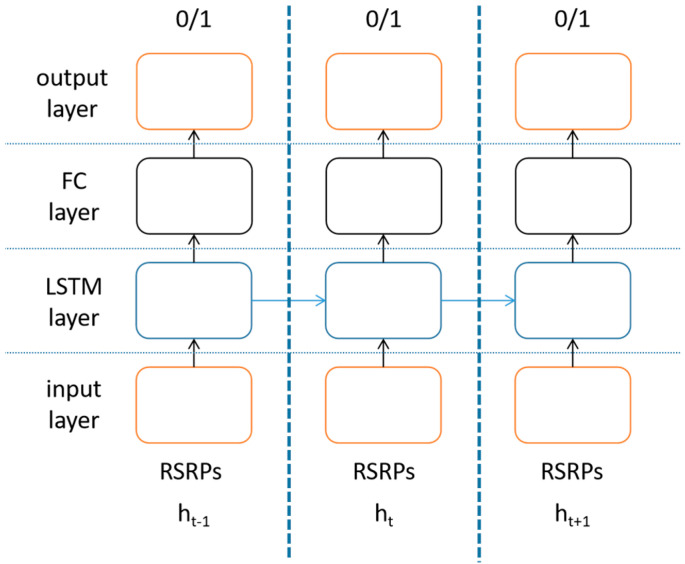
LSTM network.

**Figure 6 sensors-23-02191-f006:**
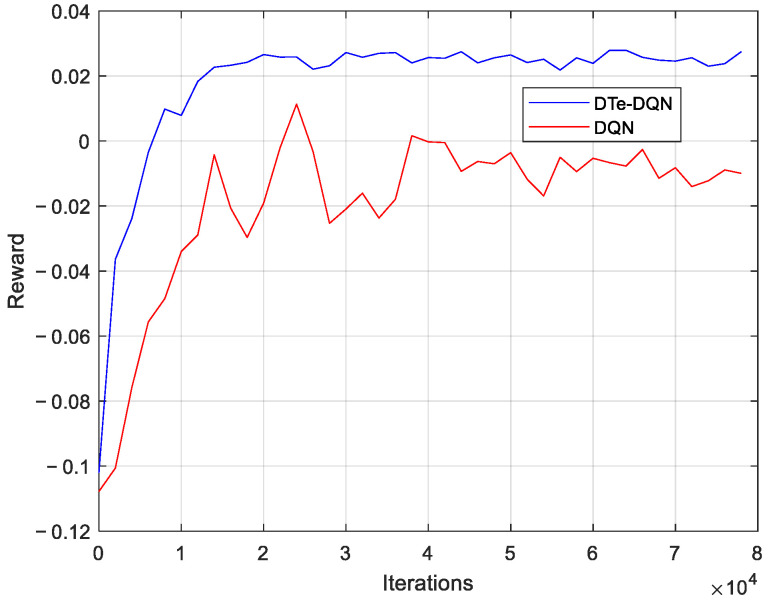
Average reward iterative curves.

**Figure 7 sensors-23-02191-f007:**
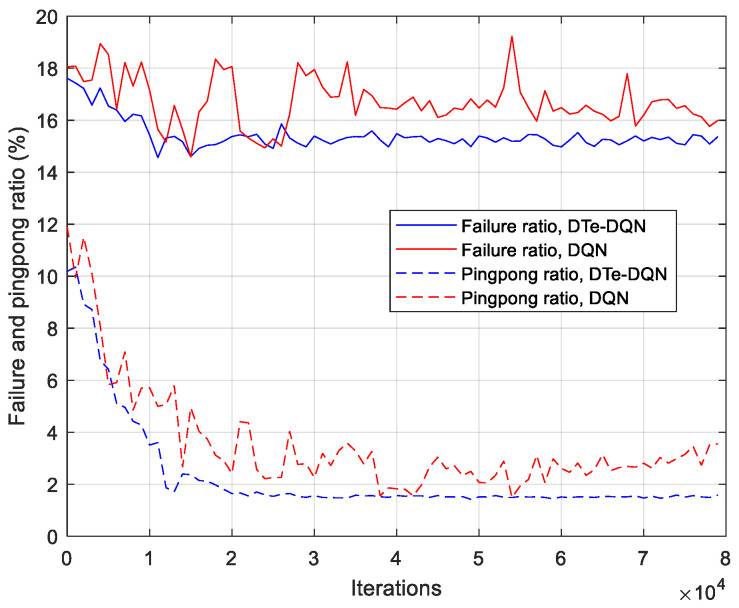
Average handover failure rate and ping-pong rate iterative curves.

**Figure 8 sensors-23-02191-f008:**
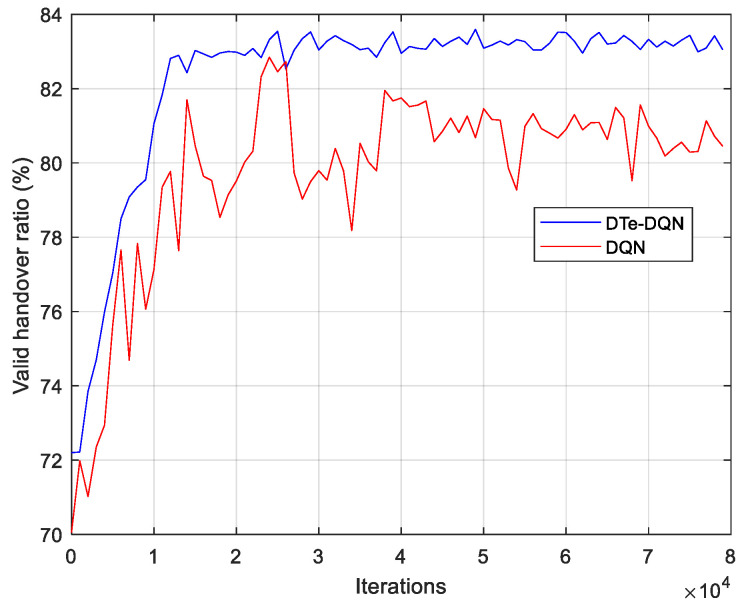
Average effective handover rate iterative curves.

**Figure 9 sensors-23-02191-f009:**
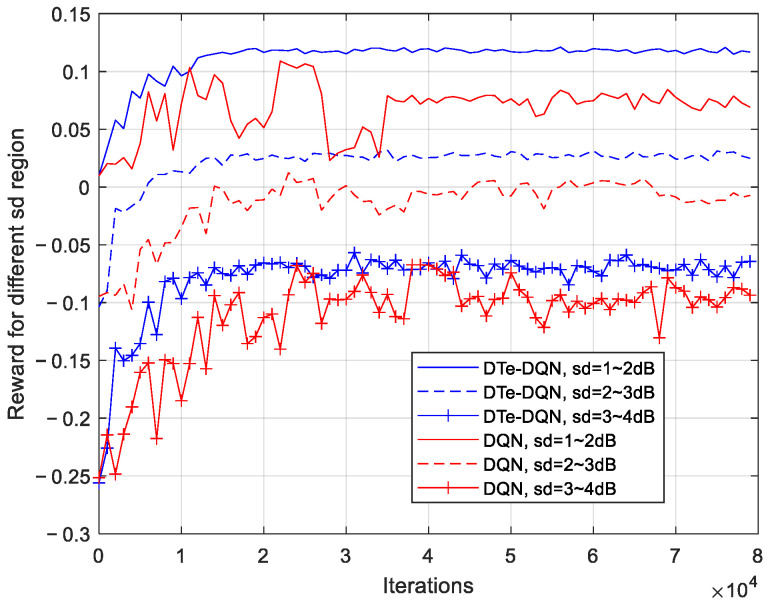
Average reward iterative curves for each fading interval.

**Figure 10 sensors-23-02191-f010:**
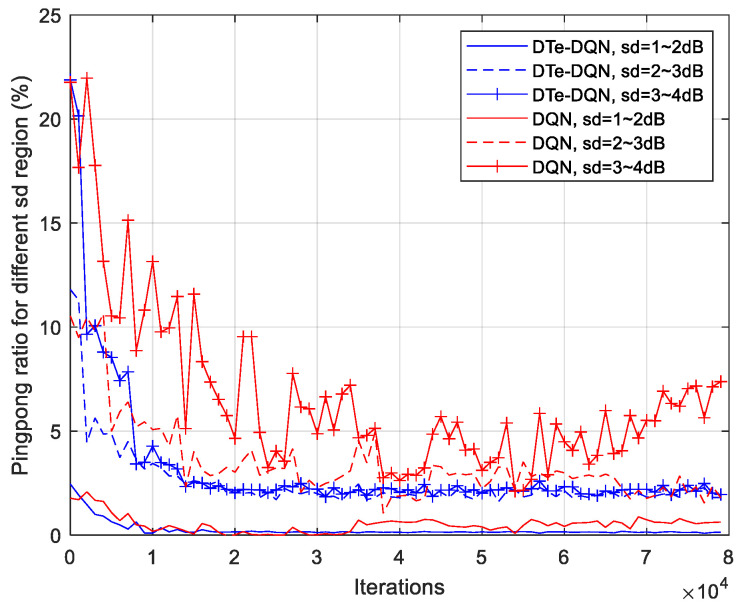
Average handover ping-pong rate iterative curves for each fading interval.

**Figure 11 sensors-23-02191-f011:**
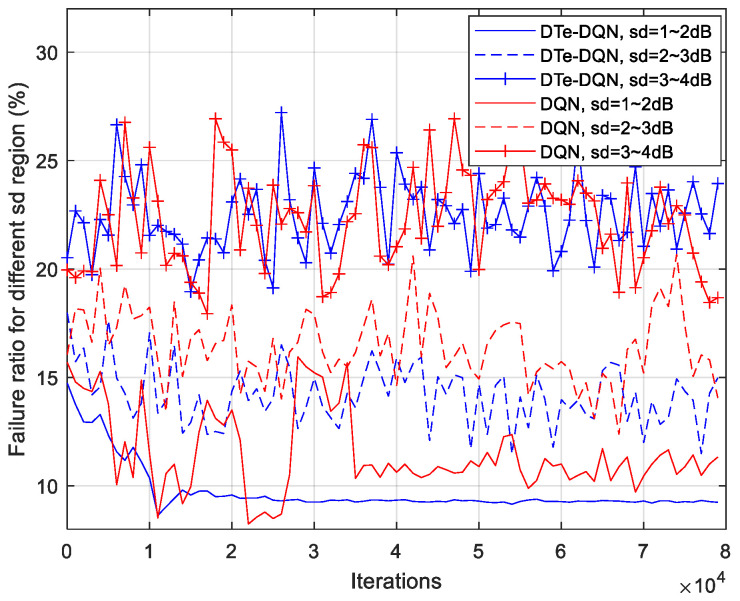
Average handover failure rate iterative curves for each fading interval.

**Figure 12 sensors-23-02191-f012:**
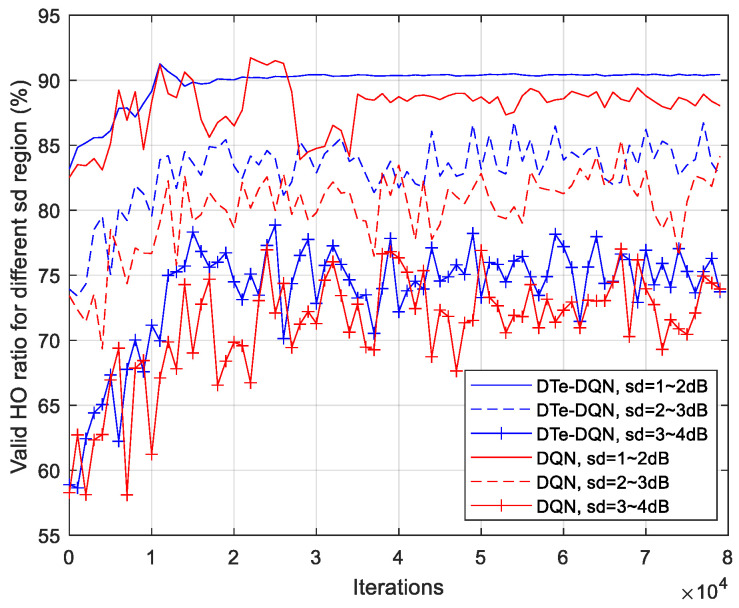
Average effective handover rate iterative curves for each fading interval.

**Figure 13 sensors-23-02191-f013:**
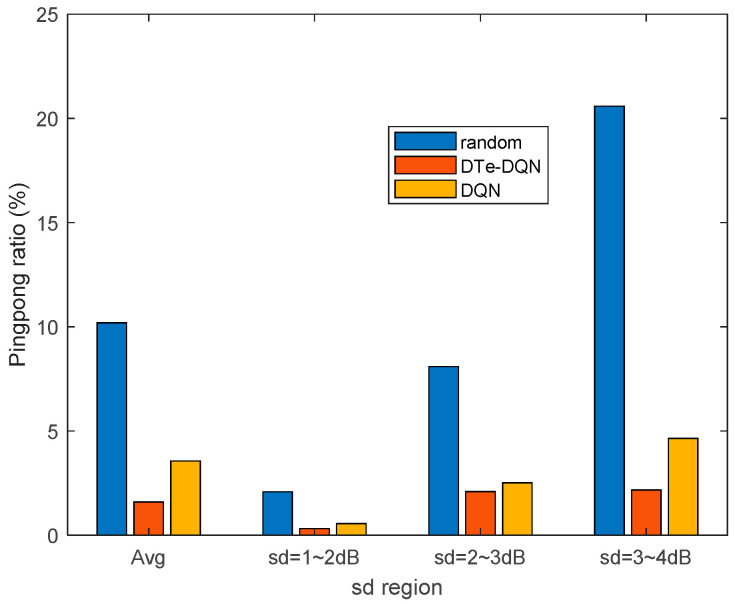
Achievable handover ping-pong rate in various cases.

**Figure 14 sensors-23-02191-f014:**
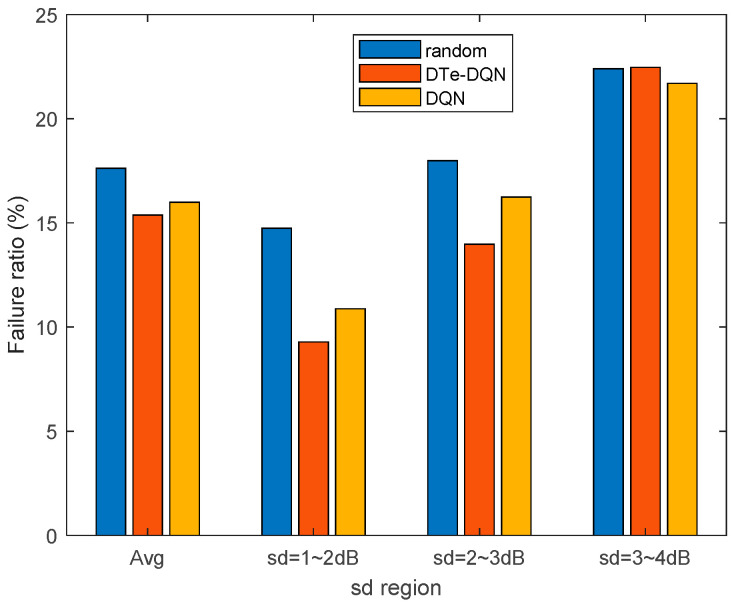
Achievable handover failure rate in various cases.

**Figure 15 sensors-23-02191-f015:**
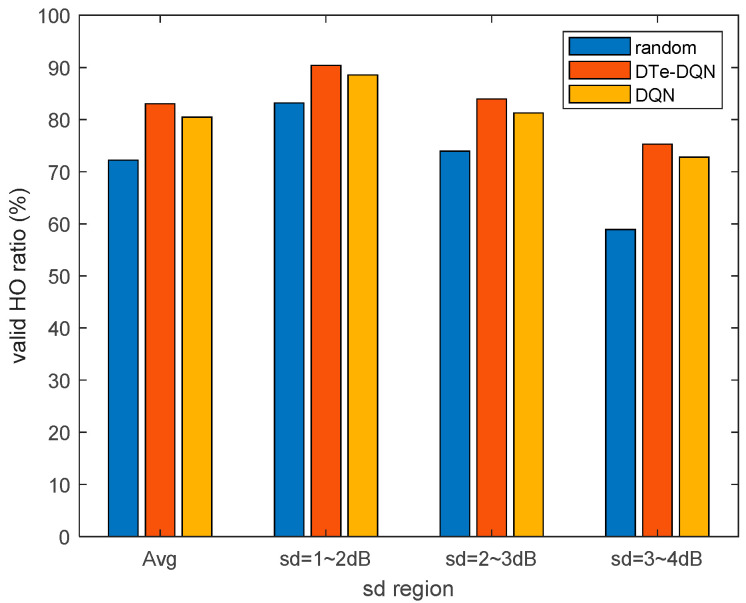
Achievable effective handover rate in various cases.

**Table 1 sensors-23-02191-t001:** Simulation and algorithm parameters.

Parameters	Configurations
$n_{\mathrm{period}}$	80,000
$n_{\varepsilon }$	500
β	0.9
Learning rate decay	Exponential decay
Optimizer	Adam
Activation	ReLu
Initial learning rate	0.001
Inter-site distance	50 m

## Data Availability

Not applicable.

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
