# Peer review of "A Reinforcement Learning Handover Parameter Adaptation Method Based on LSTM-Aided Digital Twin for UDN"

_sensors, 2023, doi:10.3390/s23042191_

Round 1
Reviewer 1 Report
The manuscript describes a DQN method that dynamically selects handover parameters according to wireless signal fading conditions using LSTM to build a digital twin. The authors claim that the proposed method is enhanced and has s a faster convergence speed than the ordinary DQN method, and at the same time achieves an average effective handover ratio increase of 2.7%. The proposed method achieved better performance in different wireless signal fading intervals. The work claims following research contributions: 1) a real-time handover parameters election method using DQN, 2) An enhanced DQN method based on digital twin; 3) Digital twin with LSTM network as the core is established.
There are some main issues with the manuscript. The first most is the writing style of the document. A major part of the document is written in “future” tense which in my opinion should be avoided. It put in doubt the whole work and is result. It is not clear that either the proposed method is in working condition/deployed or it’s just an idea on which authors are working and they are trying to get results.
In the introduction section, the authors conclude many findings without referring any previous work, for instance: how they concluded that “the design of a reasonable handover strategy to reduce the number of handovers during the user's mobility and ensure user experience has become an urgent?”
How and why the authors argue that existing approaches they their performance is slow, low efficiency, poor backward compatibility, etc.?
The author used the term “TTT” at many places without declaring it properly in the document. What is an A3 event? No detail is provided for this term.
At line No. 160-164, the authors rise/drop of ping-pong and setting of parameters without giving any reference and technical information.
At line 184-187, the researchers assumed about the change in fading conditions. What is the basis of this changes? The transformation of problem optimization is described as well.
At Line No. 190-193, the authors also talked about the Q-learning performance without giving any reference or technical information.
The algorithms proposed in the work is not analyzed for their computational resource’s requirements.
At line 228-233, the claimed advantages of the proposed DQN model are not validated.
There citation style is different at many places.
Reviewer 2 Report
This paper proposes a DQN approach that select the handover parameters according to signal fading conditions. The topic of the paper is timely and it has a good contributions. However, I have the following comments:
1) The writing of the paper needs to be further improved. The author use different tenses in the same sentences, which weaken the structure of the paper. There are some typos and grammatical issues. I suggest proofreading the paper. I include some examples,
a) There are some un clear sentences, i.e.,
“ Facing the ever-increasing demand for wireless data services, the existing network structure can no longer meet the explosive growth of data services, and ultra-dense network (UDN) is an effective and low-cost solution “
“However, the researches above have the following disadvantages: for the flexibility and performance of the algorithm, the protocol stack has been modified to a large extent, which is low in efficiency, might have poor backward compatibility, and is difficult to be implemented”
b) DQN and LSTM have not been defined in the abstract.
c) The facts and statistics included in the paper should be supported by references, i.e., “The dense deployment of small base stations can shorten the distance between mobile terminals and base stations, greatly improve the link quality between base stations and terminals, enhance signal coverage, and improve system capacity.”
d) TTT should be defined in the introduction.
e) In such sentence, “[4] proposed a DQN algorithm, according to the user's 53 SINR and access rate, adjust the handover trigger timing”
you can write
“ The author in [4] studied…….”
f) Again, same tense should be used in the literature review.
g) SINR, UE, and HOM have not been defined.
h) Equations should be followed by full stops.
2) The motivation behind proposing DQN to solve the problem should be clearly provided.
3) Regarding the optimization framework in (5), is there any practical constraint for such framework?
4) The authors should justify the results presented in the figures, rather than describing them.
5) The author should provide a comparison between the proposed DQN approach and the conventional approaches to solve the handover problem, what is the advantages of using DQN.
Reviewer 3 Report
This paper proposes a DQN handover parameter optimization scheme based on LSTM-assisted digital twins. A DQN handover parameter selection method for different wireless signal fading conditions is established, an LSTM-assisted digital twin is utilized to enhance performance by improving system efficiency and convergence effect. Overall, this paper is of high quality. The analysis is sound and the results are convincing. However, there are some parts can be further polished to improve this paper:
(1) The abbreviations should be explained when they are first used. In this paper, many are not, such as TTT, Hys.
(2) Since a deep Q-learning method is applied in this paper, it is better to describe the model following the elements of MDP, e.g., states, actions, and rewards.
(3) Although digital twin is used in this paper,it is not 'a gift from Heaven'. The mechanism is not clear here. The authors should explain how to build and implement this digital twin and how to evaluate and gurantee its accuracy and availability.
Round 2
Reviewer 1 Report
The manuscript describes a DQN method that dynamically selects handover parameters according to wireless signal fading conditions using LSTM to build a digital twin. The authors claim that the proposed method is enhanced and has s a faster convergence speed than the ordinary DQN method, and at the same time achieves an average effective handover ratio increase of 2.7%. The proposed method achieved better performance in different wireless signal fading intervals. The work claims following research contributions: 1) a real-time handover parameters election method using DQN, 2) An enhanced DQN method based on digital twin; 3) Digital twin with LSTM network as the core is established.
I previously review the manuscript and raised some points. The researchers had taken this feedback into consideration and updated the document. For instance, the authors gave sufficient detail before making any conclusion. The authors also gave the answer about computational resource requirement of the proposed algorithm.
However, in several parts, the authors used the future form which should be avoided.
Author Response
Many thanks for your kind advice. We have checked and revised the tense we use in this paper.
Reviewer 2 Report
The authors have addressed my comments.
Author Response
Many thanks for your kind advice. We have checked the paper.